# Effects of High Concentration Nitrogen Gas Stunning of Pigs on the Quality Traits of Meat and Small Intestine

**DOI:** 10.3390/ani12172249

**Published:** 2022-08-31

**Authors:** Muhammad Shahbubul Alam, Dong-Heon Song, Jeong-Ah Lee, Van-Ba Hoa, Inho Hwang, Hyoun-Wook Kim, Sun-Moon Kang, Soo-Hyun Cho, Kuk-Hwan Seol

**Affiliations:** 1Animal Products Utilization Division, National Institute of Animal Science, Rural Development Administration, Wanju-gun 55365, Korea; 2Department of Animal Science, Jeonbuk National University, Jeonju 561756, Korea; 3Department of Animal Resources Science, Kongju National University, Yesan 32439, Korea

**Keywords:** nitrogen gas stunning, pigs, quality, loin, small intestine

## Abstract

**Simple Summary:**

To satisfy consumer demand, it is necessary to ensure the quality of the meat and small intestine of animals. For consideration of animal welfare and to obtain an adequate quality of the meat and small intestine, animals need to be stunned to minimize anxiety, pain, distress, or suffering. Electric stunning and exposure to a high concentration of carbon dioxide (CO_2_) are common stunning methods in the meat industry. However, both methods have some limitations. Pale color and a higher tendency of ecchymosis is the common feature in electric-stunned pigs’ meat. Exposure to higher concentration of carbon dioxide induces high aversion, irritation of nasal mucosal membranes, and severe respiratory distress causing hyperventilation and breathlessness in pigs. Hypercapnia, hyperglycemia, lactic acidosis, and an increased hematocrit also occur in pigs. To reducing the negative impacts of CO_2_, inert gases (i.e., Argon and N_2_) are sometimes mixed with CO_2_. But, the amount of Argon in the atmosphere is very negligible (0.93%) and Argon has a high cost in the market. To maintain hypercapnic–hypoxia and improve stability, a higher concentration of N_2_ has sometimes been mixed with CO_2_ and used in pig stunning. But, no trial has used only high concentration of nitrogen in the stunning of animals, while maintaining a hypoxic condition (less than 2% O_2_ in atmosphere). Our study involved conducting a trial on electricity, CO_2_ (80%), and high concentrations of nitrogen (98%) in the stunning of pigs and comparing the meat and small intestine quality traits. Findings show that the meat and small intestine quality of N_2_ (98%)-stunned pigs was favorable compared to the other treatment groups.

**Abstract:**

The objective of present study was to investigate the feasibility of utilizing only high concentration of nitrogen gas in the stunning of pigs and its effects on the quality traits of the meat and small intestine.To conduct this experiment, three treatment groups were compared: (i) electric stunning (T1), (ii) CO_2_ (80%) gas stunning (T2), and (iii) N_2_ (98%) gas stunning (T3). A total of 21 standard pigs (Landrace × Yorkshire × Duroc; LYD) were collected from a commercial pig farm, randomly selecting seven pigs for each group (body weight of 104.5 to 120.6 kg). For stunning, each individual pig was separately kept in a gas chamber, after which each specific gas was used to fulfill the desired level in the pit. To obtain the desired level of concentration for each gas (N_2_ at 98% and CO_2_ at 80%), approximately 80 min and 35 min were required, respectively. It was observed that after reaching the desired level of concentration, pigs were stunned within a very short time (for CO_2_, 90 s and for N_2_, 120 s). For electric stunning, standard quality electric devices were used. After slaughtering, the meat and small intestine of each animal was collected separately and kept in a cool room where temperature was −2 °C. In the meat and small intestine, L* (Lightness) and b* (Yellowness) were high (*p* < 0.05) in the T1 and T3 groups. The T2 group showed high a* (Redness) (*p* < 0.05) values in both the meat and small intestine. A proximate composition of meat showed no significant differences except moisture. The water holding capacity (WHC), cooking loss (CL), and Warner-Bratzler shear force (WBSF) of meat were lowest in the T2 group, but not at a notable difference compared to T3. In the small intestine, L^*^ (Lightness)**,** a* (Redness), b* (Yellowness), and thickness significantly differed (*p* < 0.05) in each group, but WBSF showed no significance between the T2 and T3 groups. It is concluded that a high concentration of N_2_ gas (98%) may be considered in the stunning of pigs, and its effect on meat and small intestine is favorable.

## 1. Introduction 

In view of animal welfare at slaughter, animals must be stunned to minimize anxiety, pain, distress, or suffering [1]. Standard electric- and carbon-dioxide-based stunning are the commonly practiced methods for pigs [1,2]. It has been stated that animals can instantly become unconscious using electric stunning. But, pale-colored meat and a high prevalence of drip loss were found in electric-stunned pigs [3,4,5,6]. A higher tendency of ecchymosis may occur in electric-stunned meat [7,8], affecting a lower value of pork [9]. When introducing high concentration of CO_2_ (hypercapnia), pigs were not immediately unconscious and a pronounced aversion that ignores animal welfare was observed [10,11,12,13]. Nasal annoyance and respiratory distress was also induced in carbon-dioxide-based stunning [14].

High concentration of inert gases (argon, nitrogen) have been used in the stunning process for pigs, reported as non-aversive and not show any respiratory difficulties [10,15,16]. They can displace O_2_ in the atmosphere and create hypoxic conditions (present < 2% O_2_), which influences the reduction of O_2_ level in the blood, thus infuriating central nervous system (CNS) and causing collapse [11,17]. But, Argon has high limitations to use for commercial purposes due to its negligible presence in the atmosphere (0.93%). On the other hand, N_2_ is the main component of the air (78%) and can be used widely for commercial purposes for its low price. Higher concentration of N_2_ can be used for stunning in pigs and reportedly, it did not show aversion and respiratory distress such as with Argon [11,15,18]. For maintaining hypercapnic–hypoxia conditions and enhancing stability, higher concentration of N_2_ (70%, 80%, 85%, 90%, 92%) was mixed with CO_2_ and used in pig stunning [18,19]. But studies on high concentration of only nitrogen gas used in the stunning of animals while maintaining hypoxic conditions (less than 2% O_2_ in atmosphere) have not yet been conducted.

The aims of this experiment were to test the feasibility of using only high concentration of nitrogen gas (98%) in the stunning of pigs and its effects on the quality traits of meat and small intestine.

## 2. Materials Methods

### 2.1. Experimental Design and Facilities

To assess the feasibility of using only nitrogen gas in the stunning of pigs and to compare this method with standard electric- and high carbon-dioxide-based stunning, an experimental trial was conduct on National Institute of Animal Science (NIAS), RDA, Korea. A modern digital gas chamber (length 2200 mm × width 1000 mm × height 1350 mm) was used for gas stunning. Three stunning treatment was used (i) electricity (T1), (ii) high carbon dioxide (T2) (80%), and (iii) only nitrogen gas (T3) (98%). Twenty-one standard pigs (Landrace × Yorkshire × Duroc; LYD) were collected from the nearest commercial pig farm. Pigs were randomly selected for each of the treatment groups. The average body weight range was 104.5 to 120.6 kg.

The gas chamber was placed adjacent to the electric stunning room of the slaughter house. Pigs were supplied from a pig farm by their own vehicle (pickup van). After arrival, pigs were unloaded from the van and kept in pens by the staff of the slaughterhouse. No feed was supplied in the pens; only ad-libitrum water was supplied to drink. Pigs were kept in pens for 4–5 h. Just before 10–15 min of stunning activities, each individual pig of each specific treatment group was taken from pen to gas chamber via carrying trolley; slaughterhouse staff helped with this. The same facilities were provided for all pigs. When pigs were kept in the gas chamber, a very gentle approached was used. Then, the doors of gas chamber were tightly closed and animal conditions were strictly monitored. After a few minutes, when pigs were calm, the desired gas cylinder was opened and gas flowed into the chamber. “Gas in and out” of chamber was controlled by two flow meters. The concentration of gas inside the pit of chamber was monitored by gas detector, which was placed outside of the chamber. The animals conditions, such as movement, behavior, attitude, or any others symptoms, were strictly monitored by an internal close circuit camera (CC camera, Supreme Thermal Instrument (STI), Dasa-eup, Dalseong-gun, Daegu, Korea) and sensor, which were placed inside of the gas chamber, and connected to an externally placed computer that displayed animal conditions. Gas (CO_2_ or N_2_) flow continued into the pit of chamber until the desired level of concentration (80% or 98%) was reached. It was observed that reaching 80% concentration of CO_2_ gas in the pit of the chamber required about 35 min, but in the case of N_2_, it required about 80 min to reach at 98%. After reaching the desired concentration level, gas flow was stopped and the time (seconds) until the stunned state was reached was counted. It was found that within 90 s, pigs were completely stunned by CO_2_ (80%), and in the case of N_2_ (98%), it required about 120 s for complete stunning. For electric stunning, standard electric device was used, where pigs were stunned within 8–10 s. The duration of the stunning and slaughtering session of this experiment was 10 days.

### 2.2. Slaughtering and Sample Collection

After stunning, each pig was taken out of gas chamber as quickly as possible. For electric stunning, each stunned pig was kept on a movable table in the stunning room. One hind leg of each stunned pig (both electric- and gas-stunned) was tied with an iron chain (hock joint) and hung with a conveying elevator. Then, pigs were slaughtered by sharp knife and left for some minutes for proper bleed out. Each carcass was then passed through steam for a short time (45–60 s) for scalding. For removal of extra hair from the body, a fire was lit. Next, each pig’s head was cut and removed from its carcass. After this, the gut, liver, lung, and other soft organs were eviscerated and kept in a plastic bucket. Each carcass was then longitudinally divided into two parts by electric knife and stored in a cold room where temperature was −2 °C. Each small intestine was collected separately, cleaned and washed with tap water, and stored in the cold room. For each treatment group, seven animals (*n* = 7) and thus seven small intestines (*n* = 7) were used. For every small intestine, 150 cm was collected for fresh analysis and 150 cm for cooked. After 24 h, each carcass was transferred from the cold room to fabricating room, where the loin meat was collected. Each small intestine was also collected from the cold room after 24 h, as well as removing unnecessary fat.

### 2.3. pH Measurement

To determine the pH, around 100 g of meat from each sample (extra fat free) was first properly blended with a hand blender, and from that, 5 g sample was taken. A total of 50 mL distilled water was measured by measuring cylinder. Pre-measured meat samples and distilled water were properly mixed with a small hand blender. This mixture was then poured into a test tube and then a pH reading was taken with the help of a digital pH meter (Mettler-Toledo GmbH pH meter, Wetherill Park, NSW, Australia). Before taking the reading, the pH meter was calibrated by yellow (pH = 7), pink (pH = 4), and blue (pH = 9) color solutions. In the case of the small intestine, 3 g properly blended sample was taken and mixed with 30 mL distilled water. This mixture was taken in a test tube, where the pH reading was measured. For every sample (both meat and small intestine), the pH reading was taken 4 times.

### 2.4. Color Values

To measure color values, samples (loin meat and small intestine, Pendle Hill NSW, Australia) were bloomed in air for about 40 min. Color values were taken by digital Chroma Meter (Konica Minolta, Tokyo, Japan). Firstly, the camera and monitor of chromometer was connected by a data cable and then they were calibrated by a white plate where Y = 86.3, X = 0.3165, and y = 0.3242. In the case of the meat, for each rectangular (15 cm × 6 cm) sample, color values were measured from 6 different angles of the surface and side view. For the small intestine, for each sample, 150 cm of a fresh, clean small intestine was taken and placed longitudinally upon a plastic white board. Then, values were taken from 6 different places where the minimum distance from one place to another was 10 cm. Color values were articulated as L* (Lightness), a* (Redness), and b* (Yellowness).

### 2.5. Proximate Components

To detect the proximate components of meat, from each sample, around 250 g of meat was properly blended by hand mixer. Then, a sample was taken for analysis. A Food Scan^TM^ Lab (Foss Tech., Hillerød, Denmark) instrument was used to conduct proximate analysis. First, the food scanner was set up, warmed up, and the necessary programs were downloaded. Then, a round disc container was compacted with blended meat and placed in the food chamber of instrument under the scanner (with camera and light). After this, the door of the food chamber was closed tightly. Then, necessary directions were provided in the computer for the “meat products” category. It required about 30–40 s for one time analysis. After this, the disc was taken out of food chamber and the meat sample was reversed. For every sample, the scan was conducted 3 times. Using the Food Scan Lab instrument, protein, fat, moisture, ash, and collagen was detected.

### 2.6. Cooking Loss (CL) and Warner-Bratzler Shear Force (WBSF)

To determine cooking loss (CL) of meat, firstly, steaks (10 cm × 3.5 cm) of each meat sample were made from the loins. Then, they weighed by balance and placed in heat enduring plastic bags. The mouth of the plastic bags were folded and blocked by plastic clips. A digital precise shaking water bath, “Wise Bath SSB” (Scilab Korea Co., Ltd., Seoul, Korea) was used for cooking. The water bath was filled with lukewarm water. Then, plastic bags of meat samples were hung in water bath with the help of springs. The electric switch of the water bath was then turned on for boiling water up to 72 °C. While water boiled, a stainless steel cover was used for covering the water bath. To detect the interior temperature of each meat steak, a thermo-recorder was used. When the interior temperature reached 70 °C, meat samples were taken out of the water bath and placed in an ice water basin for cooling for around 30 min. After this, cooked meat samples were removed from the plastic bags and wiped with tissue paper to absorb the extra water. Then, the cooked meat samples were weighed and the cooking lose was calculated by formula.

After calculating cooking loss, the same cooked steaks were used to determine WBSF. From each steak, 4 to 5 cores were made (length 2.5 to 3.0 cm and diameter 1.25 cm) using a metal corer (diameter 1.25 cm). Cores were made according to muscle fiber direction. Then, these cores were cut using an Instron Universal Testing Machine (Model 4465, Instron Corp, Norwood, MA, USA) where the speed was 200 mm/min and the load cell was 40 N; a 1 cm distance was maintained from one cut to another. Values were recorded on a computer.

In the case of the small intestine, WBSF was determined in both fresh and cooked samples. For the fresh measurement, from each small intestine, a 50 cm long, fresh and clean sample was taken and directly cut using the Instron machine; a minimum 5–7 cm distance was maintained for the cutting interval. For cooked samples, firstly, a 50 cm long clean sample was taken from each small intestine and placed in heat enduring plastic bags. The mouth of each plastic bag was sealed with steal clips. After this, sample bags of each small intestine were placed in a “Wise Bath SSB” water bath, which was pre-filled with lukewarm water. Sample bags were hung in water bath with springs. A boiling temperature in the water bath was set at 71.3 °C. After covering the water bath with a stainless steel cover, the switch was turned on. Around 40 min was required to reach the desired temperature. After cooking, the sample bags of each small intestine were shifted from the water bath to an ice water basin for cooling. Around 30 min later, the sample bags were removed from the cold water basin and samples were taken out of the bags. Then, samples were then wiped with tissue to remove surface water and then they were ready for measuring the WBSF values. Cooked samples were cut using the Instron machine at 5–7 cm intervals and values were recorded on a computer.

### 2.7. Water-Holding Capacity (WHC)

To determine the WHC of meat, firstly, meat samples (additional fat free) were grinded to a very fine form, where fibers or coarse particles were almost absent. Then, grinded meat samples were properly checked and if any fibers, fat, or coarse particles were present, they were smoothly removed. After this, 0.51 g of good quality grinded meat was taken from each sample and inserted into a pre-weighed ultra-centrifugal tube. Then, these tubes were heated in a “Wise Bath SSB” water bath at 80 °C for 20 min. After this, they were shifted to a centrifugal machine and centrifuged at 4 °C for 10 min (speed–2000 rpm). After centrifugation, tubes were taken out and left for 10 min at a rising room temperature (25 °C). After this, cooked samples were weighed (including the ultra-centrifugal tube) and the WHC of meat was calculated by formula. For each sample, the above procedure was performed twice and the average value was taken.

### 2.8. Thickness of Small Intestine

Thickness was measured both in fresh and cooked samples of the small intestine. For the thickness measurement of each fresh sample, a 50 cm long clean and fresh sample of each small intestine was retrieved. Then, samples were longitudinally spread on white plastic board and the thickness was measured using a digital caliper scale. The reading was taken from 6 different locations of the small intestine. Distance from one location to another was a minimum of 5–7 cm. After measuring, these fresh samples were used for cooking. For the thickness measurement of each cooked sample, first, the fresh samples were placed in heat enduring plastic bags and then the mouth of the bags were folded and sealed with steel clips. After this, these bags were placed in a water bath, which was pre-filled with lukewarm water. The desired boiling temperature of the water bath was set at 71.3 °C. Around 40 min later, the desired temperature was reached. Then, the sample bags were shifted to an ice water basin for cooling. After 30 min, the samples were taken out of the bags and wiped with tissue paper to absorb the extra water. Then, the cooked samples were spread on a plastic board and thickness was measured using the caliper scale; 6 readings were taken and from one location to another the distance was a minimum of 5–7 cm. All of the readings were recorded.

### 2.9. Statistical Analysis

A Statistical Analysis System (SAS) package (Cary, NC, USA, 2007) was used for data analysis in this experiment. Means, standard errors of mean (SEM), and p-value were calculated for all the treatment groups. Duncan’s multiple range test was also used. Significant difference was denoted at *p* < 0.05.

## 3. Results

### 3.1. p^H^-24 h and Color Value of Stunned Pigs Meat and Small Intestine

The p^H^-24 h and color values of the stunned pigs meat and small intestine are shown in Table 1. The results elucidate that the p^H^-24 h value of both the meat and small intestine were significantly distinct in the middle of different conduct groups; it was comparatively high in the T1 group and low in the T2 group.

For the color value of meat, L* (Lightness) had a significant difference between T1 and T2 groups, and the T3 group was not significantly affected. For a* (Redness), the T2 value was comparatively higher than other groups and significantly different. For b* (Yellowness), the T1 and T2 groups were significantly different, and the T3 group was not significantly different from T1 and T2. In the case of the small intestine, a noteworthy dissimilarity was observed in all color values (L*, a*, and b*) of both fresh and cooked conditions among the treatment groups. The L* (Lightness) and b* (Yellowness) values of the small intestine (both fresh and cooked) were relatively higher in both the T1 and T3 groups compared to the T2 group, such as in meat. However, in the case of a* (Redness) value, it was vice versa; a* (Redness) showed a higher value in the T2 group that was significantly different from the others. In cooked small intestines, the L* (Lightness) and b* (Yellowness) values were comparatively higher compared to the fresh condition.

### 3.2. Proximate Components of Stunned Pigs Meat

Proximate components of stunning pork are presented in Table 2. For the proximate composition- protein, fat, ash, and collagen showed no significant dissimilarity in different treatment group, exception moisture where T1 was significantly differ from T2 and T3. Protein content was comparatively high in the T1 group and fat content was high in the T3 group.

### 3.3. Water-Holding Capacity (WHC), Cooking Loss (CL), and Warner-Bratzler Shear Force (WBSF) of Stunned Pigs Meat

The WHC, CL, and WBSF levels of stunned pig meat are demonstrated in Table 3. In the case of WHC, a significant difference (*p* < 0.05) was found between the electric (T1) and gas stunning (both T2 and T3) groups but not between the T2 and T3 groups. No noteworthy dissimilarity in cooking loss (CL) was observed among all treatment groups. Comparatively high cooking loss (CL) was seen in the T1 group. The WBSF value was high in the T1 group with a significant difference (*p* < 0.05) compared to the other two groups, and there was no significant difference between the T2 and T3 groups.

### 3.4. Thickness and WBSF of Stunned Pigs Small Intestine (Both Fresh and Cooked)

As shown in Table 4, the thickness of both the fresh and cooked small intestine had a significant difference (*p* < 0.05) among all treatment groups. In both the fresh and cooked condition, a highest thickness value was observed in the T1 group and lowest in the T2 group. The thickness of the cooked small intestine of all treatment groups showed higher value than fresh small intestine. In WBSF, the fresh small intestine of all treatment groups provided a high value compare to the cooked samples. A high value of both the fresh and cooked samples was seen in the T1 group compared to the other treatment groups. In the fresh samples, a notable dissimilarity was found between the electric (T1 group) and gas stunning groups (T2 and T3), and in cooked samples, T2 was significantly different from the T1 and T3 groups.

## 4. Discussion

### 4.1. p^H^-24 h and Color Value of Stunned Pigs Meat and Small Intestine

The pH of the meat and small intestine is one of the most important indicators of improper stunning. Several researchers used blood samples to detect the stunning stress of swine. The highest concentration of blood lactate was produced by exposure to 80% CO_2_ for 70 seconds in the stunning of hogs, creating the greatest stress [20]. The blood p^H^ was decreased in piglets by using 70% CO_2_ in stunning while performing castration (surgery), where the blood glucose and lactate value was high [21]. Lactic acidosis and hyperglycemia indicate stress before the slaughter of an animal. According to Becerril-Herrera et al. [22], due to exposing high concentrations of CO_2_ (80%) for pigs (duration 60 s), hypercapnia, hyperglycemia, lactic acidosis and an increased hematocrit were observed. Blood lactate concentration is negatively correlated with p^H^ [23]. Glycogen breaks down due to short term stress and produces lactic acid, which causes a lower value of p^H^ [24]. Claudia et al. [25] mentioned that the p^H^ value of electrically stunned pig meat was higher than CO_2_- and N_2_O + CO_2_-stunned pig meat. Becerril-Herrera et al. [22] also found that the blood p^H^ of CO_2_-stunned pig meat (70%) was lower than those exposed to electrical stunning. Muscle pH decline occurs due to temperature, glycogen, and lactate contents. Channon et al. [8] found that pH levels that reduced the speed of electric-stunned pigs were comparatively quicker than CO_2_ stunning. The p^H^ range of standard pig meat and small intestine is 5.6 to 5.7 and 6.7 to 7.5, respectively. The present study shows that the p^H^ value of all stunned pig meat and small intestine was below the standard. This indicates that all groups of pigs were stressed during stunning. It is also considerable that the p^H^ value of the T2 group’s meat and small intestine was comparatively lower compared to the other groups. This indicates that the T2 group was more stressed than the other groups.

To satisfy consumer demand and acceptance, the color of the meat and small intestine is an important indicator of commercial meat industry. Several factors are responsible for the color value of the meat and small intestine. The meat color of animals depends on abundance of myoglobin, the chemical situation of pigments, and its physical characteristics [26,27]. Color value L* compliantly aligns with illustration tint surveillance amid a* which is accountable for the 69% inconsistency in the cherry red color [27,28]. Color value a* (Redness) is associated with the content of pigment, oxidation situation [26,27,29], and fiber types of the muscle [30]. The intramuscular fat content and redox condition of meat is related to color value b* (Yellowness) [27,29,31,32]. For electric stunning, L* values of the *Longissimus thoracis* muscle was reportedly higher and ascribed as paler meat [8]. This brightness augmentation may be caused by the spreading more light on the free water of the meat cell surface, as a result of protein denaturation [33]. According to Channon et al. [8], CO_2_-stunned pigs showed less Lightness in the muscle than those exposed to electric stunning. The meat and small intestine having higher Lightness (L*) and lower Redness (a*) is seen as comparatively pale in color. In the present study, the T1 group showed high Lightness and less Redness, the T2 group showed less Lightness and high Redness, and position of the T3 group was intermediate between the T1 and T2 groups, indicating that the meat quality of the T3 group was comparatively better than the other two groups.

### 4.2. Proximate Components of Stunned Pigs Meat

The quality of meat is a vital factor that satisfies consumer demand; it basically depends on nutrition value, physical appearance, freshness, and eating excellence of the meat itself [34,35]. Feeding, genetics, age, and gender can act as imperative functions in developing the meat quality of animals. However, regarding proximate composition in the present study, all components, such as protein, fat, ash, and collagen (but not moisture), showed no significant difference among all treatment groups. The moisture of the T1 group was comparatively high (74.57%) and significantly different (*p* < 0.05) from the other groups. The protein content was high in the T1 group (23.24%) and fat was high in the T3 (4.02%) group. Gye-Woong Kim [36] found that standard pig content is 22.6 to 25.10% for protein and 2.23 to 2.91% for fat. The protein content in the present study was in the standard range and the fat content was greater than the standard in the T2 and T3 groups. It may indicate a difference in the animals themselves or in each gas stunning effect, but this is not clear to us. Some researchers mentioned that meat with lower protein always have high fat contents [37]. The crude ash content in the present study (1.5 to 1.53%) was slightly higher than previous reports. Reportedly, the range of ash content in rib-eye and semi-tendinous muscles was 0.77 to 1.32% in cattle [37], and in standard pigs, it was 0.97 to 1.12% [36]. The slaughtering method could alter the mineral content of meat. Hafiz, Hassan, Nazmi, and Manap [38] found that halal (neck cut) and Chinese slaughtering in a broiler increased the mineral content in broiled meat. Stunning and slaughtering plays a vital role in modifying the muscular metabolism to convert muscle to meat via a series of biochemical and mechanical mechanisms [39].

### 4.3. Water-Holding Capacity (WHC), Cooking Loss (CL), and Warner-Bratzler Shear Force (WBSF) of Stunned Pigs Meat

The WHC is a vital feature for maintaining the saleable final weight of meat and meat products. A low percentage of WHC in meat would fail to fulfill export qualities and consumer demand, resulting in a huge loss in the economy [40,41]. The desirable market demand for meat is adequate marbling with high WHC. Several researchers reported that different slaughtering methods have no effect on the WHC of cattle [42], and goat meat [43,44]. Genetics and feeding supplements (minerals and vitamins D) help to improve the firmness of the cell membrane structure that increases the WHC of meat [45]. Reportedly, pre-slaughter stunning rapidly breaks down muscle glycogen and produces enlarged amounts of blood lactate that ultimately influence lower p^H^ values, decrease WHC, and cause tougher meat [24,46]. Van der Wal, Engel, and Reimert [47] observed stress conditions 1 min just before slaughter with a higher concentration of CO_2_ and found lower WHC and a decreasing quality of meat. Bond, Can, and Warner [48] reported that an increased amount of adrenaline was found in the blood due to stress, which enhanced water loss from meat. Different stunning methods (electric and gas) significantly effect (*p* < 0.05) the WHC of lamb meat at 7 and 14 days of storage, but not immediately [49]. Linares et al. [50] found that the WHC of gas (CO_2_ 80%)-stunned lamb was lower than electric stunning. The present study also demonstrated the same findings, where the T2 group content had a significantly lower (*p* < 0.05) WHC (63.08%) than T1. T3 was not significantly affected, with values between the other two groups. Due to inhalation of higher concentrations of CO_2,_ a severe breakdown of cells in lamb influenced the expulsion of a huge amount of water from the cells and subsequently decreased the WHC of the meat [51]. As per a report by Vergara and Gallego [52], a common tendency of gas-stunned lamb was to release more water from the meat.

Cooking loss (CL) of meat is caused by the contraction of muscle fibers and intramuscular connective tissue during applied heat for a certain amount of time in water. The intensity of cooking loss depends on the temperature and the appliance used for cooking. However, pre-slaughter stunning and slaughtering methods may affect cooking loss. According to Azad Behnan Sabow et al. [53], electric-stunned goats showed higher (*p* < 0.05) cooking loss than non-stunned goats at 7 and 14 d postmortem. Onenc and Kaya [42] and Linares et al. [54] also observed increased CL in electric-stunned cattle and lamb at the 1st and 2nd week’s postmortem, respectively. A faster or slower p^H^ decline rate also a vital factor for CL. A faster p^H^ decline rate showed higher cooking loss in electric-stunned animals [55]. An increased pale-colored meat of broiler and turkey demonstrated higher CL [56,57]. Electric-stunned lamb showed significantly elevated cooking loss compared to CO_2_-stunned animals at 72 h postmortem [54]. All previous reports indicate that electric-stunned meat of different animals showed higher cooking loss compared to gas stunning or no stunning, and the initial 1–2 days have no significant differences among them. The present study also agrees with this statement, where the T1 group showed higher cooking loss, and among them, no significant differences were observed.

Warner-Bratzler shear force (WBSF) is a vital indicator of meat that influences consumers’ intake contentment [58]. Several factors control WBSF of meat, such as heritability (gene), stress, handling, chilling system, cooking situation, and ageing time [59,60]. WBSF is positively related to the age of the animal; it increased in older animals compared to younger ones [61,62]. Due to the increased aging time of meat, myofibrillar proteins degraded by endogenous proteases that provided a low value of WBSF [63,64]. This report is also supported by other studies, [50,53] working with lamb and goat, respectively. Different stunning methods affect the WBSF value of meat. In other studies, electric-stunned meat showed a higher WBSF value than non-stunning [52] and CO_2_ gas stunning meat [49]. According to Dransfield’s [65] report, electric-stunned animals showed a higher WBSF value due to reducing the activity of calpain. Reportedly, the number of present enzymes and the muscle p^H^ controlled the activity of calpain [66]. On the other hand, CO_2_ affects WBSF values in meat. Generally, in exposure to higher concentrations of CO_2_ for animal unconsciousness by stunning, this gas is highly soluble in meat and can remain at prominent levels in the tissues as residue [67], thus lowering WBSF and increasing the tenderness of the meat. According to Veeramuthu and Sams [68] and Vergara, Linares et al., [69], a lower WBSF value was observed in poultry and lamb compared to electric-stunned meat. In the present study, the T1 group showed the highest WBSF value (*p* < 0.05) compared to the T2 and T3 group. No significant difference between the T2 and T3 groups was observed, where the value of the T3 group is slightly higher.

### 4.4. Thickness and WBSF of Stunned Pigs Small Intestine (Both Fresh and Cooked)

The small intestine of a pig is an important digestive and absorptive organ, where a variety of nutrients are absorbed into the body for further metabolism activities. The morphological structure of the small intestine, enzyme activities, and nutrient transfer are directly related to digestion and absorption [70]. Genetics, nutrition, and feeding habit, age, sex, disease, bacterial load, and parasites may affect the thickness of small intestine. Different stressors (transport, handling, and pre-slaughter stunning) having any effect or not on the thickness of pigs’ small intestine have not yet been studied, and the same follows for the effects of electric or gas stunning (CO_2_ or N_2_) on the small intestine. In the present study, the thickness of pigs’ small intestine (both fresh and cooked) showed a significant difference among the different treatment groups, where T1 showed highest and T2 was lowest. The thickness value of cooked small intestine (in all treatment groups) was higher than fresh.

Stunning methods can affect the WBSF value of the small intestine. Generally, electric-stunned animals showed a higher WBSF values compared to gas (CO_2_)-stunned animals. During the exposure of higher concentration of CO_2_ for animal unconsciousness, a huge amount of CO_2_ was absorbed in the meat and small intestine via inhalation and stayed in the cells at an elevated level. After this, during cooking, when the perimysium was reconstituted and converted to gelatin [71], this gas combined with the triple helix of collagen and accumulated to weak places, creating apertures and cracks and causing a lower value of WBSF [72]. The present study showed the lowest value of WBSF (*p* < 0.05) in T2 (CO_2_) compare to others and agreed with the previous mechanism.

## 5. Conclusions

It is concluded that high concentration of only nitrogen gas (98%) can be used in the stunning of pigs. According to the results, there is no more of an adverse effect on the meat and small intestine quality of only nitrogen gas (98%)-stunned pigs than of standard electric- and carbon-dioxide-based stunning (80%). So, it could be used as a valuable alternative method of stunning animals.

## Figures and Tables

**Table 1 animals-12-02249-t001:** Effect of electric and gas stunning (CO_2_-80% & N_2_-98%) on p^H^-24h and Color value of pigs meat and small intestine (*n* = 7).

Items	Stunning Treatment			SEM	*p*-Value
	T1	T2	T3		
p^H^-24h					
Meat	5.61 ^a^	5.45 ^c^	5.56 ^b^	0.018	<0.001
Small intestine (Fresh)	6.73 ^a^	6.49 ^c^	6.63 ^c^	0.023	<0.001
Color value					
Meat					
L* (Lightness)	57.04 ^a^	53.30 ^b^	55.10 ^ab^	0.625	0.041
a* (Redness)	5.08 ^b^	7.15 ^a^	5.91 ^b^	0.267	0.002
b* (Yellowness)	4.17 ^a^	2.61 ^b^	3.42 ^ab^	0.208	0.004
Small Intestine (Fresh)					
L* (Lightness)	64.86 ^a^	50.73 ^c^	58.86 ^b^	1.373	<0.001
a* (Redness)	10.28 ^c^	16.56 ^a^	13.12 ^b^	0.589	<0.001
b* (Yellowness)	8.87 ^a^	2.28 ^c^	3.76 ^b^	0.569	<0.001
Small Intestine (Cooked)					
L* (Lightness)	73.09 ^a^	61.68 ^c^	68.32 ^b^	1.082	<0.001
a* (Redness)	6.26 ^c^	15.52 ^a^	10.44 ^b^	0.873	<0.001
b* (Yellowness)	11.06 ^a^	8.50 ^b^	8.81 ^b^	0.235	<0.001

T1 = Electric stunning; T2 = CO_2_ (80%) gas stunning; T3 = N_2_ (98%) gas stunning. ^a, b, c^ Dissimilar superscript letters in same row means significant distinction (*p* < 0.05). *n* = Number of animals or small intestines for each treatment group. SEM = Standard error of mean.

**Table 2 animals-12-02249-t002:** Effect of electric and gas stunning (CO_2_-80% and N_2_-98%) on proximate components of pigs meat (*n* = 7).

Items	Stunning Treatment			SEM	*p*-Value
	T1	T2	T3		
Moisture (%)	74.57 ^a^	72.80 ^b^	72.67 ^b^	0.292	0.105
Protein (%)	23.24	23.14	22.90	0.132	0.625
Fat (%)	2.17	3.79	4.02	0.328	0.216
Ash (%)	1.50	1.50	1.53	0.025	0.849
Collagen (%)	0.88	0.83	1.00	0.063	0.483

^a, b^ Dissimilar superscript letters in same row means significant distinction (*p* < 0.05) and lack of letters means non-significant distinction. *n* = Number of animals for each treatment group.

**Table 3 animals-12-02249-t003:** Effect of electric and gas stunning (CO_2_-80% and N_2_-98%) on WHC, CL, and WBSF of pigs meat (*n* = 7).

Items	Stunning Treatment			SEM	*p*-Value
	T1	T2	T3		
WHC (%)	66.91 ^a^	63.08 ^b^	64.98 ^ab^	0.699	0.075
Cooking loss (%)	25.71	23.49	23.91	0.601	0.291
WBSF (kg/cm^2^)	3.99 ^a^	3.06 ^b^	3.15 ^b^	0.159	0.021

^a, b^ Dissimilar superscript letters in same row means significant distinction (*p* < 0.05) and lack of letters means non-significant distinction. *n* = Number of animals for each treatment group.

**Table 4 animals-12-02249-t004:** Effect of electric and gas stunning (CO_2_-80% and N_2_-98%) on thickness and WBSF of pigs small intestine (both fresh and cooked) (*n* = 7).

Items	Stunning Treatment			SEM	*p*-Value
	T1	T2	T3		
Thickness (mm)					
Fresh small intestine	1.85 ^a^	1.12 ^c^	1.36 ^b^	0.060	<0.001
Cooked small intestine	2.58 ^a^	1.75 ^c^	2.36 ^b^	0.087	<0.001
WBSF (kg/cm^2^)					
Fresh small intestine	13.09 ^a^	11.11 ^b^	11.74 ^b^	0.207	0.002
Cooked small intestine	7.07 ^a^	6.26 ^b^	6.79 ^a^	0.109	0.006

^a, b, c^ dissimilar superscript letters in same row means significant distinction (*p* < 0.05). *n* = Number of small intestines in each treatment group.

## Data Availability

Interested persons can obtain data from the corresponding author on request.

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
