# Peer review of "Effects of High Concentration Nitrogen Gas Stunning of Pigs on the Quality Traits of Meat and Small Intestine"

_animals, 2022, doi:10.3390/ani12172249_

Round 1

Reviewer 1 Report

This paper is on a subject that has an increasing interest in the area of animal welfare. The authors compared the use of nitrogen-only-based stunning methods for pigs to standard electric and to carbon dioxide-based stunning methods for pigs at the slaughterhouse. The results of this study show that the nitrogen-only stunning method could be used as a valuable alternative stunning method, in view of the fact such a system does not negatively affect meat quality. 

Even though the aim of the study is clear, there are several flaws in the material and methods, discussion, and conclusion sections that make this manuscript unsuitable to be published in its current form.

Unfortunately, the manuscript is not properly formatted, and the absence of continued numbered lines makes the reviewing particularly complicated.

Here the authors can find a list of points for consideration:

Introduction

-There is a lack of references in the first part of the introduction, I believe the authors can arrange this section including appropriate references.

-‘High market price’: il would it be interesting to unravel the reason why the market price is high

-‘It was time-consuming due to the slightly lower relative density air’: this sentence is not clear and it must be changed, what did the authors mean by time-consuming?

Material and methods

Experimental facilities

In my opinion, this chapter needs to be completely re-written. The design of the structures lacks a clear description, as well as it is not clear how the animals were moved from the pens to the stunning chambers or who did these operations.  Moreover, were the pigs raised in the same facilities or not? What kind of measures/clinical signs were used to evaluate animal conditions?

I don’t understand also the meaning of the sentence ‘for each gas treatment, it required 3 days, did the author refer to the overall duration of the experiment?

Discussion

4.1

-‘pH of the meat and small intestine is the important indicator’

I reckon the pH is an important indicator of inappropriate stunning but it is not the only one. I reckon that the authors should support this concept with more evidence, or change the sentence.

-‘Lactic academia’: Again, this sentence must be checked and modified.

Conclusion

The conclusion should summarize the content and the scientific value of the manuscript, while in this case, this section appears like a bar list of the results obtained. 

Author Response

Manuscript No. 1848245; Journal – animals

Title: Effects of High Concentration Nitrogen Gas Stunning to Pig on the Quality Traits of Meat and Small Intestine

Date: 22.08.2022

Dear Reviewer,

Have a nice time. Hope that you are doing well. We appreciate you for your precious time in reviewing our paper and providing valuable comments. The authors have carefully considered the comments and tried our best to address every one of them. We hope the manuscript after careful revisions meet your high standards. The authors welcome further constructive comments if any.

Here I have attached  point-by-point responses in ‘Blue color'.

Sincerely yours

Muhammad Shahbubul Alam

First Author

Ph.D Fellow, Chonbuk National University

& Research Assistant, NIAS, RDA, Korea

Cell: +82-010-4395-0727; email: [email protected]

Reviewer 2 Report

For consideration of animal welfare and obtaining good quality meat and small intestine, better ways to stun the animals should be explored. In this manuscript, the authors tried to test the feasibility of using only high concentration of nitrogen gas in stunning of pigs and its effect on quality traits of meat and small intestine. The experiment is appropriated designed and the results are clearly presented. I would like to recommend this work if the authors can address the following:

1.     The number of small intestine should be indicated in Materials Methods part.

2.     Repeat number (n) of each parameter should be noted in each table.

3.     About the conclusion: Although most parameters of quality traits were not significant different in this work, the authors concluded T3 (N2-98%) is better than the other two groups mostly due to the meat color. However, there is no significant difference in L* a* b* between T1 and T3. Therefore, I don’t think we should conclude T3 is best for quality traits of meat and small intestine unless the author can provide more data. Actually, the purpose of this paper is just to explore the feasibility of 98% nitrogen gas in stunning of pigs and its effect on quality traits of meat and small intestine. From the data, I think the author can conclude N2 gas (98%) can be used in stunning of pigs since there is no more adverse effect than T1 and T2 group.

Author Response

Manuscript No. 1848245; Journal – animals

Title: Effects of High Concentration Nitrogen Gas Stunning to Pig on the Quality Traits of Meat and Small Intestine

Date: 22.08.2022

Dear Reviewer,

Have a nice time. Hope that you are doing well. We appreciate you for your precious time in reviewing our paper and providing valuable comments. The authors have carefully considered the comments and tried our best to address every one of them. We hope the manuscript after
careful revisions meet your high standards. The authors welcome further constructive
comments if any.

Below we provide the point-by-point responses in ‘Blue’.

Sincerely yours

Muhammad Shahbubul Alam

First Author

Ph.D Fellow, Chonbuk National University

& Research Assistant, NIAS, RDA, Korea

Cell: +82-010-4395-0727; email: [email protected]

Round 2

Reviewer 1 Report

I would like to thank the authors for addressing all the comments provided in the first round of the revision process. In my opinion, the manuscript can be published in its current form.